# The Prognostic Value of the GNRI in Patients with Stomach Cancer Undergoing Surgery

**DOI:** 10.3390/jpm13010155

**Published:** 2023-01-13

**Authors:** Qianqian Zhang, Lilong Zhang, Qi Jin, Yongheng He, Mingsheng Wu, Hongxing Peng, Yijin Li

**Affiliations:** 1Liyuan Hospital, Tongji Medical College, Huazhong University of Science and Technology, Wuhan 430062, China; 2Department of General Surgery, Renmin Hospital of Wuhan University, Wuhan 430060, China; 3Central Laboratory, Renmin Hospital of Wuhan University, Wuhan 430060, China; 4Department of Neurology, The First Hospital of Jilin University, Changchun 130000, China; 5Department of Colorectal and Anorectal Surgery, Hunan Hospital of Integrated Tradmonal Chinese and Western Medicine (Hunan Academy of Traditional Chinese Medicine Affiliated Hospital), Changsha 410006, China

**Keywords:** geriatric nutritional risk index, stomach cancer, surgery, meta-analysis, prognosis

## Abstract

Malnutrition often induces an adverse prognosis in cancer surgery patients. The elderly nutrition risk index (GNRI) is an example of the objective indicators of nutrition-related risks. We performed a meta-analysis to thoroughly examine the evidence for the GNRI in predicting the outcomes of patients undergoing stomach cancer surgery. Eligible articles were retrieved using PubMed, the Cochrane Library, EMBASE, and Google Scholar by 24 October 2022. The clinical outcomes were overall survival (OS), cancer-specific survival (CSS), and post-operative complications. A total of 11 articles with 5593 patients were included in this meta-analysis. The combined forest plot showed that for every unit increase in the preoperative GNRI score in patients with stomach cancer, their postoperative mortality was reduced by 5.6% (HR: 0.944; 95% CI: 0.933–0.956, *p* < 0.001). The pooled results also demonstrated that a low GNRI was correlated with poor OS (HR: 2.052; 95% CI: 1.726–2.440, *p* < 0.001) and CSS (HR: 1.684; 95% CI: 1.249–2.270, *p* = 0.001) in patients who underwent stomach cancer surgery. Postoperative complications were more likely to occur in patients with a low GNRI, as opposed to those with a high GNRI (OR: 1.768; 95% CI: 1.445–2.163, *p* < 0.001). There was no evidence of significant heterogeneity, and the sensitivity analysis supported the stability and dependability of the above results. the GNRI is a valuable predictor of long-term outcomes and complications in stomach cancer patients undergoing surgery.

## 1. Introduction

Gastric cancer (GC) remains a particularly lethal cancer with the fourth highest fatality and the fifth highest incidence rate worldwide. East Asian countries have the highest incidence of gastric cancer, accounting for more than half of the reported patients [1]. Even after curative surgery, the prognosis of a significant number of GC patients remains poor. GC patients often have inadequate oral intake because of multiple cancer-associated symptoms, including obstruction, anorexia, nausea, and generalized fatigue [2]. Malnutrition is common in GC patients because of their increased metabolic demands, nutrient loss, and inadequate oral intake [3,4,5], which is the main risk factor that leads to perioperative complications [6,7]. Therefore, it is very important to evaluate the nutritional status of GC patients before surgery to optimize their prognosis.

The geriatric nutrition risk index (GNRI) is a nutritional parameter that involves the ratio of serum albumin level to current weight and ideal healthy weight, which is objective and simple compared with other parameters [8]. Compared with the serum albumin level or body mass index alone, the GNRI is thought to be a more accurate predictor of nutrition-related outcomes in aging populations [9]. The formula used to calculate the GNRI is as follows: GNRI = (1.489 × albumin, g/L) + (41.7 × present/ideal body weight, kg) [8]. Since the GNRI is easily applied in clinical practice, it is widely used to assess the nutritional status of various patients. A recent study suggested that a lower GNRI is associated with a poor prognosis in patients with esophageal cancer [9]. 

To date, several retrospective studies have analyzed the association between the GNRI and prognosis and perioperative complications in GC patients undergoing surgery. However, systematic evaluations of whether preoperative GNRI values can effectively predict the outcome of surgical treatment for GC patients have not been carried out. Therefore, in this study, we verified the impact of the GNRI on the prognosis of GC patients.

## 2. Methods

### 2.1. Literature Search Strategies

The Preferred Reporting Items for Systematic Reviews and Meta-Analyses (PRISMA) statement was used in this meta-analysis [10]. The protocol for this meta-analysis is available on PROSPERO (CRD42022369645). On 24 October 2022, PubMed, EMBASE, and Cochrane Library were searched using the following keywords: “Geriatric nutritional risk index”, “GNRI”, “Stomach Neo-plasms [Mesh]”, “Stomach Neoplasm”, “Stomach Cancer”, “Gastric Neoplasm”, “Gastric Cancer”, “Cancer of the Stomach”, “Cancer of Stomach”. The language of the studies was restricted to English. Using Google Scholar, we verified the grey documents without indexes in the above-mentioned database. In addition, we screened references that met the inclusion criteria.

### 2.2. Inclusion and Exclusion Criteria

The detailed inclusion criteria are as follows: patients with GC who underwent surgery, patients whose surgical prognosis was evaluated by research, and patients who supplied information on at least one of the outcomes of interest (overall survival (OS), cancer-specific survival (CSS), and postoperative complications). Reviews, conference abstracts, case reports, letters, and comments were excluded. If there was an overlap of patient groups in the study, we only chose the study with the most comprehensive data and the most rigorous method. 

### 2.3. Data Extraction and Quality Assessment 

The author, publication year, study region, study period, sample size, number of male and female patients, age of patients, surgical method, cut-off, and results were the primary subjects of data extraction. The quality of the observational studies was evaluated using the Newcastle–Ottawa Scale (NOS) score [11]. High-quality literature was indicated by a score below six. Two authors double-checked each of the aforementioned processes, and a senior author resolved any discrepancies.

### 2.4. Statistical Methods

Stata 15.0 was used to conduct the statistical analysis. The chi-squared test was used to determine the statistical heterogeneity. A fixed effect model was utilized when *p* > 0.1 and I^2^ 50% showed low heterogeneity; otherwise, the random-effect model was applied. To investigate the potential confounding factors in this meta-analysis, sub-group analyses were conducted. The tests of Egger and Begg were employed to evaluate publication bias. If there was a considerable publication bias, we changed the findings using the trim-and-fill technique [12]. To test the stability of the findings, a sensitivity analysis that separately excluded each study from the analysis was carried out. A *p* value of 0.05 was used to determine the significance for all the two-sided *p* values.

## 3. Results

### 3.1. Characteristics of Studies

A total of 11 studies that involved 5593 patients were included in this meta-analysis [13,14,15,16,17,18,19,20,21,22,23]. The PRISMA flow diagram is provided in Figure 1. Specifically, 83 irrelevant records were excluded after the screening of titles and abstracts. Following this, the full texts of the remaining 19 articles were further assessed. Three of these articles [24,25,26] were included in the multicenter study by Toya et al. [15] and were, therefore, excluded. GC patients with cachexia (with or without surgery) were included by Ruan et al. and were, therefore, excluded [27]. After excluding 2 unrelated studies and 2 conference abstracts, 11 articles were ultimately included [13,14,15,16,17,18,19,20,21,22,23].

The main characteristics of the studies included are shown in Table 1. A total of 10 studies were performed in Japan, whereas 1 study was conducted in Korea (Table 1). The details of the specific hospitals where the patients were recruited for each study can be found in Appendix A [13,14,15,16,17,18,19,20,21,22,23]. Three studies regarded GNRI scores as continuous variables, while eight studies reported the cut-off point of the GNRI to range from 85.7 to 98 (Table 1). Notably, 1551 patients underwent gastric endoscopic submucosal dissection (ESD), and 4042 patients underwent curative gastrectomy (Table 1). The NOS scores for 11 articles ranged from 6 to 8, which represented a low risk of bias (Table 1).

### 3.2. GNRI and Overall Survival

In total, 9 articles that involved 4948 patients explored the association between the GNRI and OS in GC patients undergoing surgery. Of these, 6 studies with 3714 patients classified patients into high and low groups using cut-off values. The pooled HR was 2.052 (95% CI: 1.726–2.440, *p* < 0.001), implying that a low GNRI raised the death risk by 105.2% (Figure 2A). Since there was no evidence of significant heterogeneity, a fixed-effects model was used (I^2^ = 0.0%, *p* = 0.633).

In addition, 3 articles with a total of 1234 participants considered the GNRI score as a continuous variable to explore its relationship with OS in GC patients. As shown in Figure 2B, a fixed-effects model was utilized (I^2^ = 45.9%, *p* = 0.158). The combined forest plot demonstrated that for every unit increase in the GNRI score in GC patients, their postoperative mortality was reduced by 5.6% (HR: 0.944; 95% CI: 0.933–0.956, *p* < 0.001).

### 3.3. GNRI and Cancer-Specific Survival

The relationship between the GNRI and CSS was also examined using prognostic data from 3 studies that involved 1960 participants. No significant heterogeneity was observed in the included studies (I^2^ = 0.0%, *p* = 0.953, Figure 3A), so a fixed-effects model was used. We found that patients with a low GNRI had worse CSS than those with a high GNRI (HR: 1.684, 95% CI: 1.249–2.270, *p* = 0.001, Figure 3A).

### 3.4. GNRI and Postoperative Complications

A connection between the GNRI and postoperative complications in GC patients was observed in a total of 6 studies that involved 3565 individuals. Hisada et al. assessed ESD-related complications based on the Common Terminology Criteria for Adverse Events version 5.0, with a CTCAE grade of ≥2 being considered as an adverse event [28]. In the remaining five studies, according to the Clavien Dindo classification, postoperative complications were categorized as a grade ≥ II [29]. As shown in Figure 3B, the pooled results demonstrated that postoperative complications were more likely to occur in patients with a low GNRI, as opposed to those with a high GNRI (OR: 1.768; 95% CI: 1.445–2.163, *p* < 0.001). No heterogeneity was found in the studies (I^2^ = 0.0%, *p* = 0.512), and a fixed-effects model was applied to this analysis.

### 3.5. Subgroup Analysis of OS and Postoperative Complications

We subsequently performed a subgroup analysis by correcting for the impact of publishing year, treatment, sample size, GNRI cut-off value, and definition of complications. The results revealed that the GNRI was an independent prognostic factor that affected the OS and postoperative complications of the patients in all the subgroups (Figure 4 and Figure 5 and Appendix A).

### 3.6. Publication Bias

The publication bias was verified by Begg’s and Egger’s tests. We confirmed that there was no evidence of publication bias for OS (Egger’s test: *p* = 0.825; Begg’s test: *p* = 0.707) or CSS (Egger’s test: *p* = 0.436; Begg’s test: *p* = 1.000) across the studies. Notably, the publication bias for postoperative complications was found by Egger’s test (Egger’s test: *p* = 0.004; Begg’s test: *p* = 0.452). Next, the trim-and-fill method was used to calculate the number of missing studies on postoperative problems. By factoring in the missing hypothesis studies, the combined OR was recalculated, but was not substantially different (HR: 1.592, 95% CI: 1.332–1.902; *p* < 0.001, Appendix A). As a result, the publication bias had little impact, and the outcome was quite stable.

### 3.7. Sensitivity Analysis

We used the leave-one-out method to perform a sensitivity analysis to determine how each study might affect the meta-analysis. As shown in Figure 6A, the pooled HR for OS did not significantly change after excluding one study at a time and ranged from 1.999 (95% CI: 1.611–2.481, after omitting the study by Sugawara et al. 2021) to 2.137 (95% CI: 1.782–2.564, after omitting the study by Tsuchiya et al. 2022). Similarly, the pooled OR for postoperative complications was not significantly different in the sensitivity analysis (Figure 6B). The overall OR ranged from 1.700 (95% CI: 1.369–2.112, after omitting the study by Furuke et al. 2021) to 2.168 (95% CI: 1.627–2.890, after omitting the study by Sugawara et al. 2021). From the above, we can conclude that our results are stable and reliable.

## 4. Discussion

This study aims to verify the predictive significance of the GNRI in GC patients treated with surgery, and the pooled data demonstrated that a higher GNRI was strongly related to longer OS and CSS and lower postoperative complications in GC patients. Furthermore, our findings held stable even after the sensitivity analysis and subgroup analysis were used to detect potential confounders, suggesting that a lower preoperative GNRI is an independent indicator of a poorer prognosis for surgery in GC patients. To the best of our knowledge from a comprehensive search of the literature, this is one of the only meta-analyses to thoroughly explore the impact of the GNRI on the prognosis of GC patients undergoing surgery. As a highly accessible indicator in clinical practice, preoperative assessment of patients’ GNRI and nutritional interventions for patients with a higher GNRI (e.g., >98) can be extremely helpful in improving the prognosis of these patients.

Malnutrition is detrimental to the immune system and is associated with inflammation and cachexia, which significantly increase the risk of postoperative complications [30,31], diminish the effectiveness of chemoradiotherapy, and increase the likelihood of adjuvant therapy adverse effects [32,33,34], all of which are directly related to the patient’s prognosis. Therefore, several biomarkers were developed, including the PNI [35] and CONUT [36], to detect patients who were malnourished. However, these indices were lacking in value for older patients, due to limitations in usual weight estimation [37]. Next, the GNRI was proposed and was used as an age-specific indicator to assess the nutritional status of elderly patients. Surprisingly, recent research has suggested that the GNRI may have better predictive value than nutritional assessment in many diseases, such as heart failure [38,39], hemodialysis [40], and patients undergoing surgery for various malignancies (for example, colorectal cancer [41], pancreatic cancer [42], gallbladder cancer [43], hepatocellular carcinoma [44], and esophageal cancer [45]). A recent study by Chen et al. also revealed that the GNRI can be used as a promising alternative to the Global Leadership Initiative on Malnutrition (GLIM) and is the best option for the perioperative management of patients with rectal cancer [46]. Compared with other types of cancer, the nutritional metabolism disorder of gastric cancer patients is more serious and specialized. Because the stomach is one of the main organs for digesting food and plays an important role in the nutrition and metabolism of the body [47,48], it is necessary to study the nutrition of gastric cancer patients for the prognosis of gastric cancer [49].

Cancer cachexia is a complex pathological disorder caused by the interaction of complex factors, such as inflammation, hypermetabolism, changes in neurohormones, and metabolic disorders [50,51,52]. It is characterized by clinical symptoms such as muscle atrophy, weight loss, fatigue, and anorexia [53]. It is reported that the vast majority of patients with advanced cancer will suffer from cachexia, which is not only a common and persistent pain factor for patients with advanced cancer, but also seriously affects the quality of life of patients and the effect of radiotherapy and chemotherapy [54]. In the case of cancer cachexia, the nutrition intake and metabolism of the body are more difficult, thus leading to a vicious circle [55]. Consistent with our study, a previous study indicated that good nutritional status and nutrition-centered comprehensive treatment can help to improve patients’ health by reducing their nausea and vomiting symptoms [56].

Aging and unhealthy diet are examples of the risk factors for gastric cancer [57,58]. It has been demonstrated that a short interval between lunch and dinner and a lack of exercise after dinner are the risk factors for gastric cancer, and the synergistic effect of these two risk factors is positively related to age, so the risk of gastric cancer in people over 55 years old is high [59,60]. In addition, the research on this topic suggests that vitamin supplementation is strongly related to a decrease in the incidence rate of gastric cancer [61,62]. It may be attributed to the inhibition of redox reactions by vitamin C and E, which clear the accumulation of reactive oxygen species (ROS) induced by oxidative stress in the process of gastric cancer [63,64]. Overall, the GNRI can be a promising predictor of poor outcomes in cancer patients undergoing surgery, so we concentrated on how it affected GC. We synthesized the existing evidence to confirm that the GNRI can be a valid predictor of poor outcomes in GC patients undergoing surgery. This study offers evidence-based support for the clinical use of the GNRI in the preoperative assessment of GC patients. Additionally, the critical value range for the GNRI for most of the included studies was 92–98, which may provide some reference value for determining the critical value of the GNRI in clinical applications.

However, this analysis has several limitations. Firstly, the analysis only included retrospective cohort studies, rather than well-designed randomized controlled trials (RCTs), which possibly limited its statistical power. Secondly, there is a lack of uniformity in the cut-off values of the GNRI across the studies, and the aggregated survival results may deviate from the actual values. Finally, since no Western studies were included and the patients were all from Asia, there may have been some selection bias in the patients’ ethnicity, and the conclusions may not be practical for patients of other ethnicities. Thus, to confirm and update our conclusion, more high-quality studies with sizable sample sizes, particularly multicenter RCTs, are urgently required. At the same time, these studies should also include patients of different ethnicities and explore the optimal cut-off values to more precisely guide clinical practice for the benefit of patients.

## Figures and Tables

**Figure 1 jpm-13-00155-f001:**
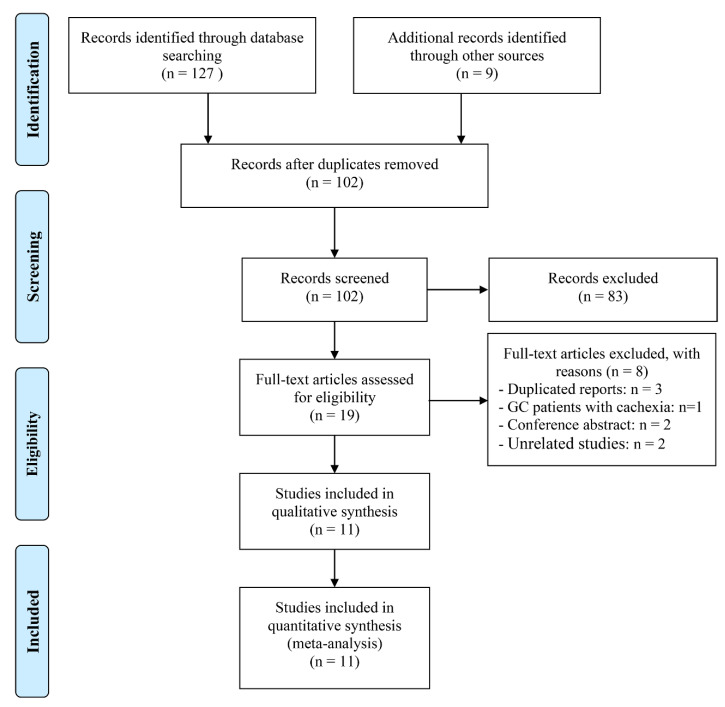
The flow diagram of identifying eligible studies.

**Figure 2 jpm-13-00155-f002:**
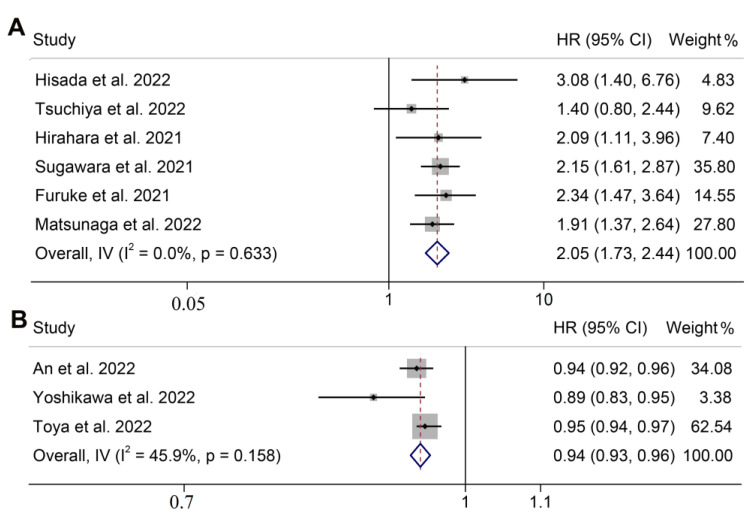
Meta-analysis of overall survival. Forest plot of the GNRI (dichotomous variable) in relation to overall survival (**A**). Forest plot of GNRI (continuous variable) in relation to overall survival (**B**). HR, hazard ratio; CL, confidence interval; GNRI, geriatric nutrition risk index [13,14,15,16,17,18,19,20,21].

**Figure 3 jpm-13-00155-f003:**
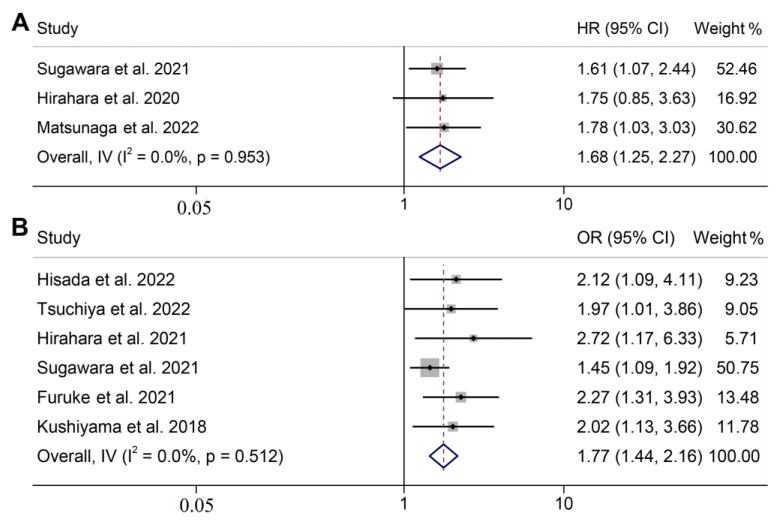
Forest plot of the GNRI in relation to cancer-specific survival (**A**) and postoperative complications (**B**). HR, hazard ratio; OR, odds ratio; CL, confidence interval; GNRI, geriatric nutrition risk index [14,16,17,19,20,21,22,23].

**Figure 4 jpm-13-00155-f004:**
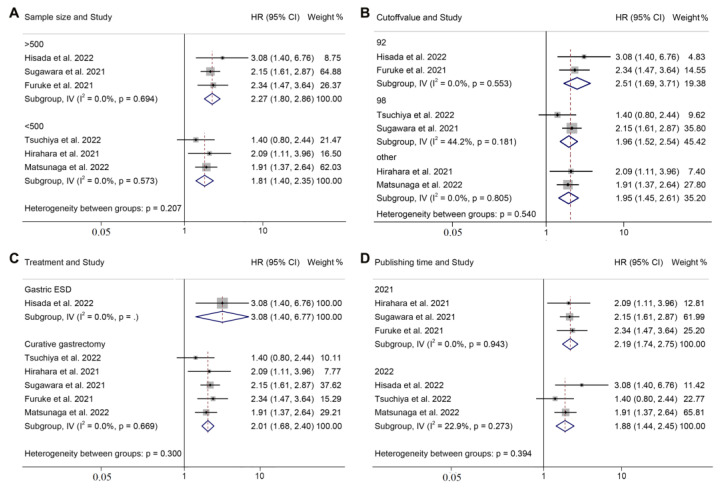
Subgroup analysis of overall survival based on sample size (**A**), cut-off value (**B**), treatment (**C**), and publishing year (**D**). ESD, endoscopic submucosal dissection; HR, hazard ratio; CL, confidence interval [14,16,17,18,19,20,21].

**Figure 5 jpm-13-00155-f005:**
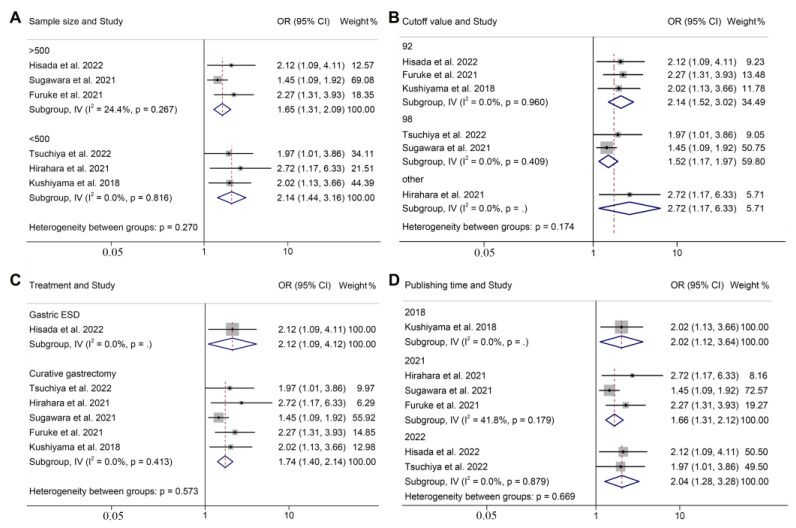
Subgroup analysis of postoperative complications based on sample size (**A**), cut-off value (**B**), treatment (**C**), and publishing year (**D**). ESD, endoscopic submucosal dissection; OR, odds ratio; CL, confidence interval [14,16,17,18,19,20,21].

**Figure 6 jpm-13-00155-f006:**
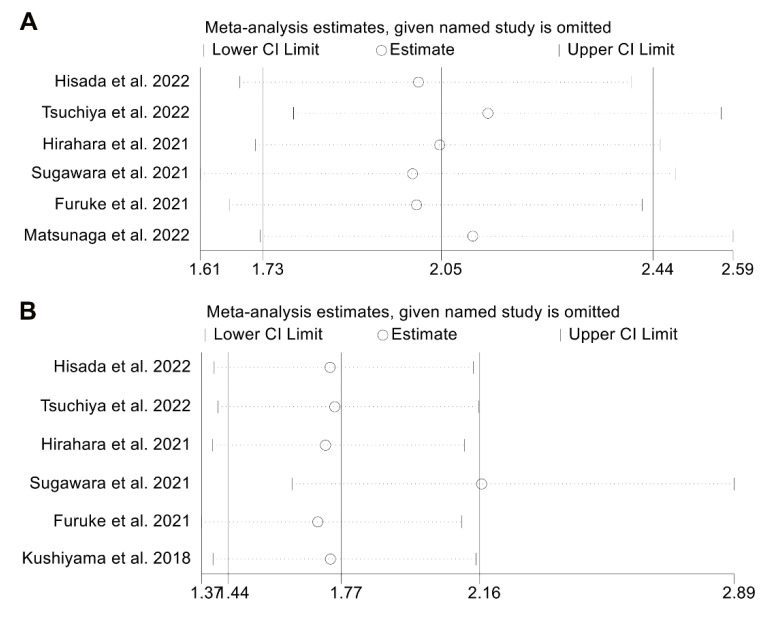
Sensitivity analysis of overall survival (**A**) and postoperative complications (**B**). CL, confidence interval [14,16,17,19,20,21].

**Table 1 jpm-13-00155-t001:** Main characteristics of the studies included.

Study	Study Region	Study Design	Study Period	Sample Size	Male/ Female	Age (Years)	Treatment	Cut-Off	Outcome	NOS Score
Toya et al. 2022 [15]	Tohoku, Japan	R	January 2002–December 2017	740	469/271	86 (85–93.0) ^a^	Gastric ESD	Continuous	OS (U)	6
Matsunaga et al. 2022 [16]	Multi-center, Japan	R	January 2005–December 2015	497	330/167	80.6 ± 4.0	Curative gastrectomy	97/95.8	OS (M), CSS (M)	8
An et al. 2022 [18]	Gangdong, Korea	R	June 2006–December 2017	450	301/149	60 (52–69) ^a^	Curative gastrectomy	Continuous	OS (M)	7
Hisada et al. 2022 [17]	Tokyo, Japan	R	January 2009–December 2019	767	559/208	75 (65–95) ^b^	Gastric ESD	92	OS (M), complications (M)	8
Yoshikawa et al. 2022 [13]	Osaka, Japan	R	January 2006–December 2020	44	30/14	86 (85–96) ^b^	Gastric ESD	Continuous	OS (U)	6
Tsuchiya et al. 2022 [14]	Yokohama, Japan	R	April 2002–December 2018	186	128/58	82 (80–93) ^b^	Curative gastrectomy	98	OS (U), complications (M)	7
Hirahara et al. 2021 [20]	Shimane, Japan	R	January 2010–December 2017	303	209/94	65–91 ^c^	Curative gastrectomy	85.7	OS (M), complications (M)	7
Sugawara et al. 2021 [19]	Tokyo, Japan	R	April 2001–December 2014	1166	816/350	25–91 ^c^	Curative gastrectomy	98	OS (M), CSS (M), complications (U)	8
Furuke et al. 2021 [21]	Kyoto, Japan	R	2008–2016	795	534/261	68 (29–89) ^b^	Curative gastrectomy	92	OS (M), complications (U)	8
Hirahara et al. 2020 [22]	Shimane, Japan	R	January 2010–December 2017	297	205/92	65–91 ^c^	Curative gastrectomy	90.9	CSS (M)	7
Kushiyama et al. 2018 [23]	Osaka, Japan	R	January 2006–December 2015	348	230/118	79.6 ± 3.8	Curative gastrectomy	92	Complications (M)	7

R: retrospective study; ESD, endoscopic submucosal dissection; OS, overall survival; CSS, cancer-specific survival; M, multivariate analysis, U, univariate analysis, ^a^ medians with interquartile ranges; ^b^ medians with ranges; ^c^ age with ranges.

## Data Availability

The data that support the findings of this study are available from the corresponding author upon reasonable request.

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
