# Peer review of "The Prognostic Value of the GNRI in Patients with Stomach Cancer Undergoing Surgery"

_jpm, 2023, doi:10.3390/jpm13010155_

Round 1

Reviewer 1 Report

Thank You for providing me with the opportunity to review this article. The manuscript before me is well-written, easy to follow and, in all, an interesting take on a previously unexplored problem, namely the assessment of the effect of geriatric nutrition risk index (GNRI) on the outcomes of gastric cancer (GC) patients. Therefore, it presents a step forward in understanding the connection between nutrition risks and health outcomes.

However, there are a couple of issues the authors would need to shed some more light on in order to make this article more clear and comprehensive.

In 2.1 the authors explain the rationale behind their search strategies, but there are no mention of language dictating the choice of studies. Were only the articles written in english included or were there some articles in other languages? Could this be a source of bias and should it, therefore, be mentioned in the limitations?

In 2.2 the authors state “When studies reported overlapping patient populations, only the study with the most comprehensive data and rigorous methods was selected.” What is considered under “the most rigorous methods”? How did the authors make a distinction between those that were or were not rigorous enough? 

Author Response

Response to Reviewer 1 Comments

Thank you for your and the reviewers' comments concerning our manuscript entitled “The prognostic value of the GNRI in patients with stomach cancer undergoing surgery”. Those comments are all valuable and very helpful for revising and improving our manuscript. We have studied the comments carefully and made corrections. The revised portions are marked in red in the manuscript. The main corrections in the manuscript and the response to your and the reviewer's comments are as follows:

  1. Point 1: In 2.1 the authors explain the rationale behind their search strategies, but there are no mention of language dictating the choice of studies. Were only the articles written in english included or were there some articles in other languages? Could this be a source of bias and should it, therefore, be mentioned in the limitations?

Response: Thank you very much for your advice. “Language restricted to English” was added.

  1. Point 2: In 2.2 the authors state “When studies reported overlapping patient populations, only the study with the most comprehensive data and rigorous methods was selected.” What is considered under “the most rigorous methods”? How did the authors make a distinction between those that were or were not rigorous enough?

Response: Thank you very much for your advice. When reporting a replicated population, we prioritised the inclusion of multicentre, larger sample size studies compared to single centre studies. As stated in our manuscript, three of these articles [24-26] were included in the multicenter study by Toya et al. [15] and were therefore excluded. Ref. 15 is a multicentre study with a larger sample size, whereas Refs. 24-26 are single-centre studies, so we chose to include Ref. 15.

  1. Toya Y, Shimada T, Hamada K, Watanabe K, Nakamura J, Fukushi D, et al. Prediction model of 3-year survival after endoscopic submucosal dissection for early gastric cancer in elderly patients aged ≥ 85 years: EGC-2 model. J Cancer Res Clin Oncol. 2022. Epub 2022/05/14. doi: 10.1007/s00432-022-04024-y. PubMed PMID: 35546359.
  2. Toya Y, Endo M, Akasaka R, Morishita T, Yanai S, Nakamura S, et al. Prognostic nutritional index is an independent prognostic factor for older patients aged ≥ 85 years treated by gastric endoscopic submucosal dissection. BMC Gastroenterology. 2021;21(1). doi: 10.1186/s12876-021-01896-1.
  3. Shimada T, Yamagata T, Kanno Y, Ohira T, Harada Y, Koike Y, et al. Predictive Factors for Short-Term Survival after Non-Curative Endoscopic Submucosal Dissection for Early Gastric Cancer. Digestion. 2021;102(4):630-9. Epub 2020/09/16. doi: 10.1159/000510165. PubMed PMID: 32932255.
  4. Toya Y, Endo M, Nakamura S, Akasaka R, Yanai S, Kawasaki K, et al. Long-term outcomes and prognostic factors with non-curative endoscopic submucosal dissection for gastric cancer in elderly patients aged ≥ 75 years. 2019;22(4):838-44. doi: 10.1007/s10120-018-00913-9.

Reviewer 2 Report

In this meta-analysis study, Zhang et al have evaluated the preoperative geriatric nutrition risk index (GNRI) in retrospectively predicting the clinical outcomes of patients who had undergone stomach cancer surgery. The evaluation has included the GNRI effect with regard to the patients’ overall survival (OS) and cancer-specific survival (CSS)/postoperative complications. Based on pooled data from 11 selected articles, with a total of 5555 patients enrolled in the respective cohorts, the authors have overall shown that higher GNRI is significantly associated with longer OS and CSS and lower postoperative complications. A sensitivity analysis of the results for finding potential confounders has further suggested the value of low preoperative GNRI as an independent indicator of a poorer prognosis for patients with gastric cancer undergoing surgery. The manuscript is easy to follow, and the results, which are presented well, are of great importance to the field. Nevertheless, improvements in the manuscript are required based on the following major and minor comments:

Major comment:

The authors state in the Discussion that “To the best of our knowledge from a comprehensive search of the literature, this is the first meta-analysis to thoroughly explore the impact of GNRI on the prognosis of GC patients undergoing surgery.” The authors are required to clarify this statement and their findings in relation to the similar published meta-analysis study by Liu et al (DOI: 10.3389/fsurg.2022.1020482) which has not been cited. Additionally, the recently published study by Chen et al (doi: 10.3389/fnut.2022.1061944) should be cited and discussed.

Minor comments:

Introduction:

1. Page 1, reference Nr 3: Please add more relevant references

Methods (page 2):

Section 2.3: Please add a reference regarding the NOS score utilized for the evaluation.

Results:

Section 3.1:

1. Please begin the section with a sentence indicating the initial number of the recorded patients (i.e. n=5555).

2. The first two sentences can be combined, e.g. "After .... were removed, resulting in 102 articles..."

3. A sentence is lacking regarding the excluded 83 records, and why they were excluded, which has resulted in the "remaining 19 articles”. Please complete the missing part and revise the sentences accordingly.

4. Please remove the dot before reference Nr 14.

5. Please reorder the exclusions shown in the "Full-text articles excluded" table in the same order as mentioned in the text (or vice versa).

Table 1:

The studies by Sugawara et al and Furuke et al are highlighted with the same color. Please revise.

Figures 2 and 3: The GNRI abbreviation can be used instead of using the full description.

Figure S2: Please indicate what exactly each axis in the graph shows, as well as what the symbols in the graph represent.

Discussion (page 9):

1.  Paragraph 2: In addition to the given reference Nr 29, please add more relevant references.

2. Line 4 from the bottom: "Malignancies [36, 37]" can be combined with the sentence which follows it.

Author Response

Response to Reviewer 2 Comments

Thank you for your and the reviewers' comments concerning our manuscript entitled “The prognostic value of the GNRI in patients with stomach cancer undergoing surgery”. Those comments are all valuable and very helpful for revising and improving our manuscript. We have studied the comments carefully and made corrections. The revised portions are marked in red in the manuscript. The main corrections in the manuscript and the response to your and the reviewer's comments are as follows:

  1. Point 1: The authors state in the Discussion that “To the best of our knowledge from a comprehensive search of the literature, this is the first meta-analysis to thoroughly explore the impact of GNRI on the prognosis of GC patients undergoing surgery.” The authors are required to clarify this statement and their findings in relation to the similar published meta-analysis study by Liu et al (DOI: 10.3389/fsurg.2022.1020482) which has not been cited.

Response: Thank you very much for your advice. We do not cite Liu et al's study because we found their results to be flawed. As stated in our manuscript, three of these articles [24-26] were included in the multicenter study by Toya et al. [15] and were therefore excluded. However, we found that the study by Liu et al. incorporated both Refs. 15 and 25. This will lead to the double inclusion of some of the population, which in turn will affect the reliability of the conclusions. For reasons of rigour, we have not cited altered literature.

  1. Toya Y, Shimada T, Hamada K, Watanabe K, Nakamura J, Fukushi D, et al. Prediction model of 3-year survival after endoscopic submucosal dissection for early gastric cancer in elderly patients aged ≥ 85 years: EGC-2 model. J Cancer Res Clin Oncol. 2022. Epub 2022/05/14. doi: 10.1007/s00432-022-04024-y. PubMed PMID: 35546359.
  2. Toya Y, Endo M, Akasaka R, Morishita T, Yanai S, Nakamura S, et al. Prognostic nutritional index is an independent prognostic factor for older patients aged ≥ 85 years treated by gastric endoscopic submucosal dissection. BMC Gastroenterology. 2021;21(1). doi: 10.1186/s12876-021-01896-1.
  3. Shimada T, Yamagata T, Kanno Y, Ohira T, Harada Y, Koike Y, et al. Predictive Factors for Short-Term Survival after Non-Curative Endoscopic Submucosal Dissection for Early Gastric Cancer. Digestion. 2021;102(4):630-9. Epub 2020/09/16. doi: 10.1159/000510165. PubMed PMID: 32932255.
  4. Toya Y, Endo M, Nakamura S, Akasaka R, Yanai S, Kawasaki K, et al. Long-term outcomes and prognostic factors with non-curative endoscopic submucosal dissection for gastric cancer in elderly patients aged ≥ 75 years. 2019;22(4):838-44. doi: 10.1007/s10120-018-00913-9.

  1. Point 2: Additionally, the recently published study by Chen et al (doi: 10.3389/fnut.2022.1061944) should be cited and discussed.

Response: Thank you very much for your advice. We read this literature carefully and we found that this study included patients with rectal cancer and could not be included in our meta-analysis. (Chen XY, Lin Y, Yin SY, Shen YT, Zhang XC, Chen KK, Zhou CJ, Zheng CG. The geriatric nutritional risk index is an effective tool to detect GLIM-defined malnutrition in rectal cancer patients. Front Nutr. 2022 Nov 15;9:1061944. doi: 10.3389/fnut.2022.1061944.)

However, it is undeniably a very interesting study. In our discussion, we have quoted from this article and added the following: “A recent study by Chen et al. also revealed that GNRI can be used as a promising al-ternative to the Global Leadership Initiative on Malnutrition (GLIM) and is the best option for the perioperative management of patients with rectal cancer [46].”

  1. Point 3: Page 1, reference Nr 3: Please add more relevant references.

Response: Thank you very much for your advice. In accordance with your suggestion, we have cited the following references.

Mariette C, De Botton ML, Piessen G. Surgery in esophageal and gastric cancer patients: what is the role for nutrition support in your daily practice? Ann Surg Oncol. 2012;19(7):2128-34. Epub 2012/02/11. doi: 10.1245/s10434-012-2225-6. PubMed PMID: 22322948.

Saunders J, Smith T. Malnutrition: causes and consequences. Clin Med (Lond). 2010;10(6):624-7. Epub 2011/03/19. doi: 10.7861/clinmedicine.10-6-624. PubMed PMID: 21413492; PubMed Central PMCID: PMCPMC4951875.

  1. Point 4: Section 2.3: Please add a reference regarding the NOS score utilized for the evaluation.

Response: Thank you very much for your advice. In accordance with your suggestion, we have cited the following references.

Stang A. Critical evaluation of the Newcastle-Ottawa scale for the assessment of the quality of nonrandomized studies in meta-analyses. Eur J Epidemiol. 2010;25(9):603-5. Epub 2010/07/24. doi: 10.1007/s10654-010-9491-z. PubMed PMID: 20652370.

  1. Point: 5: (1) Please begin the section with a sentence indicating the initial number of the recorded patients. (2) The first two sentences can be combined, e.g. "After .... were removed, resulting in 102 articles...". (3) A sentence is lacking regarding the excluded 83 records, and why they were excluded, which has resulted in the "remaining 19 articles". Please complete the missing part and revise the sentences accordingly.

Response: Thank you very much for your advice. Based on your suggestions, we have reorganised the language of this paragraph. The revised text reads as follows: “A total of 11 studies involving 5,593 patients were included in this meta-analysis [11-21]. The PRISMA flow diagram is provided in Figure 1. Specifically, 83 irrelevant records were excluded after the screening of titles and abstracts. Later, the full-texts of the remaining 19 articles were further assessed. Three of these articles [22-24] were in-cluded in the multicenter study by Toya et al. [13] and were therefore excluded. GC pa-tients with cachexia (with or without surgery) were included by Ruan et al. and were therefore excluded. [25]. After excluding 2 unrelated studies and 2 conference abstracts, 11 articles were ultimately included [11-21].”

  1. Point 6: Please remove the dot before reference Nr 14.

Response: Thank you very much for your advice. The dot before reference Nr 14 was deleted.

  1. Point 7: Please reorder the exclusions shown in the "Full-text articles excluded" table in the same order as mentioned in the text (or vice versa).

Response: Thank you very much for your advice. We have modified the order in the Figure 1.

  1. Point 8: Table 1: The studies by Sugawara et al and Furuke et al are highlighted with the same color. Please revise.

Response: Thank you very much for your advice. We have amended this issue.

  1. Point 9: Figures 2 and 3: The GNRI abbreviation can be used instead of using the full description.

Response: Thank you very much for your advice. “Figure 2 Meta-analysis of overall survival. Forest plot of the geriatric nutrition risk index (dichotomous variable) in relation to overall survival (A). Forest plot of geriatric nutrition risk index (continuous variable) in relation to overall survival (B)” was revised to “Figure 2. Meta-analysis of overall survival. Forest plot of the GNRI (dichotomous variable) in relation to overall survival (A). Forest plot of GNRI (continuous variable) in relation to overall survival (B)”.

“Figure 3 Forest plot of the geriatric nutrition risk index in relation to cancer-specific survival (A) and postoperative complications (B)” was revised to “Figure 3. Forest plot of the GNRI in relation to cancer-specific survival (A) and postoperative complications (B)”.

  1. Point 10: Figure S2: Please indicate what exactly each axis in the graph shows, as well as what the symbols in the graph represent.

Response: Thank you very much for your advice. “Theta, the effect estimate; Se_theta, the corresponding standard error; The circles represent the studies included in this meta-analysis; Boxes with circles represent additional studies of the trim and fill method.” was added.

  1. Point 11: Paragraph 2: In addition to the given reference Nr 29, please add more relevant references.

Response: Thank you very much for your advice. In response to your suggestion, we have added the following references.

  1. Kono T, Sakamoto K, Shinden S, Ogawa K. Pre-therapeutic nutritional assessment for predicting severe adverse events in patients with head and neck cancer treated by radiotherapy. Clin Nutr. 2017;36(6):1681-5. Epub 2016/11/17. doi: 10.1016/j.clnu.2016.10.021. PubMed PMID: 27847115.
  2. Tashiro M, Yamada S, Sonohara F, Takami H, Suenaga M, Hayashi M, et al. Clinical Impact of Neoadjuvant Therapy on Nutritional Status in Pancreatic Cancer. Ann Surg Oncol. 2018;25(11):3365-71. Epub 2018/08/12. doi: 10.1245/s10434-018-6699-8. PubMed PMID: 30097739.

  1. Point 12: Line 4 from the bottom: "Malignancies [36, 37]" can be combined with the sentence which follows it.

Response: Thank you very much for your advice. In response to your suggestion, we have revised the formulation of these two sentences. Revised as follows: “Surprisingly, recent research has suggested that the GNRI may have better predictive value than nutritional assessment in many diseases, such as heart failure [36, 37], haemodialysis [38] and undergoing surgery for various malignancies (for example, colorectal cancer [39], pancreatic cancer [40], gallbladder cancer [41], hepatocellular carcinoma [42], and esophageal cancer [43]).”

Reviewer 3 Report

Zhang et al. address the importance of nutrition in the prognosis of post-surgery stomach cancer. The relevance of taking into consideration of diet and nutrition for better outcomes for cancer patients including stomach cancer is needed at the time because cancer is a multifactorial disease with an unpredictable prognosis. 

However, there are substantial comments that need to be addressed for better quality of this paper.

1. The authors are encouraged to include more research data for a better conclusion.

2. It would be advisable to explore any clinical trials on diet/nutrition and prognostic values.

3. The authors may include a section and diet/nutrition and cancer cachexia.

4. The authors should include discussion in the backdrop of other cancer types along with stomach cancer for broader aspects.

5. The authors should include more references at least 100 and this help better discussion and views on nutrition and prognosis. 

6. A section can be added on Aging, Cancer and Dietary factors such as Vitamin C etc. 

6. 

Author Response

Response to Reviewer 3 Comments

Thank you for your and the reviewers' comments concerning our manuscript entitled “The prognostic value of the GNRI in patients with stomach cancer undergoing surgery”. Those comments are all valuable and very helpful for revising and improving our manuscript. We have studied the comments carefully and made corrections. The revised portions are marked in red in the manuscript. The main corrections in the manuscript and the response to your and the reviewer's comments are as follows:

  1. Point 1: The authors are encouraged to include more research data for a better conclusion.

Response: Thank you very much for your advice. We strongly agree that if more research data can be added, better conclusions will be obtained in this study. We have tried our best to retrieve the literature. Unfortunately, there is no new study that meets the standard so far. For example, we noticed a meta-analysis study by Liu et al (DOI: 10.3389/fsurg.2022.1020482), three of these articles [24-26] were included in the multicenter study by Toya et al. and were therefore excluded. However, we found that the study by Liu et al. incorporated both Refs. 15 and 25. This will lead to the double inclusion of some of the population, which in turn will affect the reliability of the conclusions. For reasons of rigor, we have not cited altered literature.

  1. Point 2: It would be advisable to explore any clinical trials on diet/nutrition and prognostic values.

Response: Thank you very much for your helpful advice. Unfortunately, at present, only the clinical trials of the elderly nutrition risk index as a prognostic indicator of small cell lung cancer patients have been reported, and there is no clinical trial of the elderly nutrition risk index as a prognostic indicator of gastric cancer patients. But there is no doubt that this will be a very good research idea.

  1. Point 3: The authors may include a section and diet/nutrition and cancer cachexia.

Response: Thank you very much for your constructive advice. As we described in the Introduction part, “Even after curative surgery, the prognosis of a significant number of GC patients remains poor. GC patients often have inadequate oral intake because of multiple cancer-related symptoms such as obstruction, anorexia, nausea, pain, and generalized fatigue [2]. Malnutrition is common in GC patients because of increased metabolic demands, nutrient loss, and inadequate oral intake [3-5], which is considered to be a main risk factor for perioperative complications [6, 7]. Thus, assessment of nutritional status in GC patients prior to surgery is essential for optimizing the outcomes”.

Besides, we added these details in the Discussion part of the revised manuscript.

Cancer cachexia is a complex pathological disorder caused by the interaction of complex factors such as inflammation, hypermetabolism, changes in neurohormones, and metabolic disorders [50-52]. It is characterized by clinical symptoms such as muscle atrophy, weight loss, fatigue, and anorexia [53]. It is reported that the vast majority of patients with advanced cancer will suffer from cachexia, which is not only a common and unrelieved pain factor for patients with advanced cancer but also seriously affects the quality of life of patients and the effect of radiotherapy and chemotherapy [54]. In the case of cancer cachexia, the nutrition intake and metabolism of the body are more difficult, thus leading to a vicious circle [55]. Consistent with our study, a previous study indicated that good nutritional status and nutrition-centered comprehensive treatment can help patients' health by reducing their nausea and vomiting symptoms [56].

  1. Point 4: The authors should include discussion in the backdrop of other cancer types along with stomach cancer for broader aspects.

Response: Thank you very much for your advice. We revised the Discussion part and marked it red. “Surprisingly, recent research has suggested that the GNRI may have better predictive value than nutritional assessment in many diseases, such as heart failure [38, 39], hemodialysis [40], and undergoing surgery for various malignancies (for example, colorectal cancer [41], pancreatic cancer [42], gallbladder cancer [43], hepatocellular carcinoma [44], and esophageal cancer [45]). A recent study by Chen et al. also revealed that GNRI can be used as a promising alternative to the Global Leadership Initiative on Malnutrition (GLIM) and is the best option for the perioperative management of patients with rectal cancer [46]”. Compared with other types of cancer, the nutritional metabolism disorder of gastric cancer patients is more serious and special. Because the stomach is one of the main organs for digesting food and plays an important role in the nutrition and metabolism of the body [47, 48]. Therefore, it is necessary to study the nutrition of gastric cancer patients for the prognosis of gastric cancer [49].

  1. Point 5: The authors should include more references at least 100 and this help better discussion and views on nutrition and prognosis.

Response: Thank you very much for your advice. We added 24 references in the revised manuscript and are expected to meet your requests.

  1. Point 6: A section can be added on Aging, Cancer, and Dietary factors such as Vitamin C etc.

Response: Thank you very much for your advice. We added these details below in the Discussion part of the revised manuscript.

Aging and unhealthy diet are one of the risk factors for gastric cancer [57, 58]. It has been demonstrated that the short interval between lunch and dinner and the lack of exercise after dinner are the risk factors for gastric cancer, and the synergistic effect of these two risk factors is positively related to age, so the risk of gastric cancer in people over 55 years old is high [59, 60]. In addition, research suggests that vitamin supplementation is strongly related to the decrease in the incidence rate of gastric cancer [61, 62]. It may be attributed to the inhibition of redox by vitamin C and E, which clear the accumulation of reactive oxygen species (ROS) induced by oxidative stress in the process of gastric cancer [63, 64].

Round 2

Reviewer 2 Report

The authors have addressed my comments and revised the manuscript accordingly. 

Reviewer 3 Report

The authors have responded to majority of suggestions.